# The diagnostic accuracy of umbilical cord procalcitonin in predicting early-onset neonatal infection

**Thi Thanh Binh Nguyen**[1,2]*, **Diep Anh Truong Thi**[1], **Quang Vinh Truong**[3], **Thi Ny Pham**[4]

**1** Department of Pediatrics, University of Medicine and Pharmacy, Hue University, Hue City, Vietnam, **2** Faculty of Pediatrics, Hue University of Medicine and Pharmacy Hospital, Hue University, Hue City, Vietnam, **3** Department of Obstetrics & Gynecology, University of Medicine and Pharmacy, Hue University, Hue City, Vietnam, **4** Faculty of Obstetrics and Gynecology, Hue University of Medicine and Pharmacy Hospital, Hue University, Hue City, Vietnam

* nttbinh.a@huemed-univ.edu.vn, nttbinh.med@hueuni.edu.vn

**Data Availability Statement:** All relevant data are within the paper and its Supporting Information files.

## Abstract

### Introduction

To determine the threshold of umbilical cord blood procalcitonin for early-onset neonatal infection diagnosis.

### Method

This prospective study was conducted on 126 neonates in the neonatal care unit of Hue University of Medicine and Pharmacy Hospital, Vietnam, from June 01, 2023 to August 31, 2024. All neonates showed signs at birth or risk factors for early-onset infection (EOI) and were divided into two groups: EOI group and non-EOI group. Umbilical cord blood samples were collected for procalcitonin analysis immediately after birth.

### Results

The median procalcitonin (PCT) levels in umbilical cord blood were significantly higher in the EOI group (0.154 ng/ml [0.092–0.197]) compared to the non-EOI group (0.097 ng/ml [0.082–0.134]; p < 0.001). Receiver operating characteristic (ROC) curve determined the optimal threshold value of PCT of 0.142 ng/ml with an AUC 0.751 (95% CI: 0.661–0.841, p<0.001) in the total population. At this cut-off, the Se, Sp, PPV, and NPV were 68.2%, 76.8%, 61.2%, and 81.8%, respectively. The optimal cut-off value for preterm neonates was 0.122 ng/ml (AUC: 0.785, 95% CI: 0.658–0.911, p<0.001) corresponding a Se of 79.2%, Sp of 74.1%, PPV of 73.1%, and NPV of 80.0%. In term group, the optimal cut-off value was 0.150 ng/ml (AUC: 0.726, 95% CI: 0.583–0.860, p<0.01), with a Se of 60.0%, Sp of 80.4%, PPV of 52.2%, and NPV of 84.9%.

### Conclusions

Umbilical cord blood PCT concentration were elevated in neonates with EOI. PCT could be a valuable marker for the early diagnosis of EOI.

**Funding:** This work was supported by research funds from Hue University (DHH 2023 – 04–202). The founders had no role in study design, data collection and analysis, decision to publish, or preparation of the manuscript.

**Competing interests:** The authors have declared that no competing interests exist.

## Introduction

Early-onset infection (EOI) is a serious and often fatal condition that afflicts neonates within the first three days of life. EOI progresses rapidly and requires immediate treatment prevent mortality [1]. Early diagnosis is essential for optimizing treatment outcomes [2–4]. In current practice, clinicians often initiate early empiric antibiotics in neonates with clinical symptoms or risk factors before obtaining evidence of infection and inflammatory markers [1]. Consequently, a large number of neonates are overdiagnosed and treated with antibiotics, leading to increased rates of drug-resistant bacteria, separation of mother and child, and increased treatment costs. Therefore, a new marker with high sensitivity and specificity, reliable for early diagnosis of EOI, and with quick results to rule out infection and avoid unnecessary antibiotic treatment is needed [2]. With the current trend, to minimize pain and blood loss for newborns, and to get results more quickly, researchers are increasingly exploring and implementing the use of umbilical cord blood as an alternative source to venous blood sampling in neonates [5, 6]. Previous studies have shown that, PCT is an early inflammatory marker with high specificity in response to severe systemic infections. It can be used for early detection, assessment of severity, and prognosis of the diseases [7, 8]. Some studies have shown that umbilical cord blood procalcitonin (PCT) concentration is a more effective marker than C-reactive protein (CRP) or white blood cells (WBCs) in identifying infections. PCT often increases early in newborns with infections [9, 10], even before they develop clear clinical symptoms. Although PCT is highly specific for infection, to avoid interference from physiological increases in PCT values after birth, it is necessary to evaluate PCT concentrations in umbilical cord blood. While there have been studies on this issue [10–12], no consensus exists on the role of umbilical cord blood procalcitonin. EOI is an infection transmitted through the maternal-fetal route (possibly through the placenta or upstream). Therefore, we hypothesize that if the newborn is exposed to the source of infection in the fetus before birth, the umbilical cord blood PCT concentration will be different, increased in neonates with EOI, and could serve as an early marker and distinguisher of EOI.

### Objective

To determine the threshold of umbilical cord blood procalcitonin for early-onset neonatal infection diagnosis.

## Subject and method

### Study design

This prospective study was conducted at the Hue University of Medicine and Pharmacy Hospital, Hue City, Vietnam, from June 01, 2023 to August 31, 2024.

### Subjects

**Inclusion criteria.** All neonates exhibited signs at birth or risk factors for early-onset infection (EOI) and measured PCT in umbilical cord blood immediately after birth. All neonates were monitored for at least the first 72 hours after birth.

**Exclusion criteria.** The family did not consent to participate in the study. The neonates were transferred to the other hospital within the first 72 hours after birth.

A total of 126 neonates was included in the final analysis.

**Variables and data collection.** Data collection: Risk factors for EOI were assessed based on NICE 2021 guideline [13] (prolonged time of rupture of membranes, maternal fever, spontaneous preterm birth before 37 weeks of gestation, mother with chorioamnionitis, maternal

streptococcus group B infection...) were collected before birth. During study period, we followed nearly 2765 newborns born in our hospital, assessing all risk factors for EOI in each pregnancy and monitoring early-onset infections. From this cohort, we obtained a final sample size of 126 cases, including 44 cases of EOI. Information about the newborns including gestational age, gender, birth weight, type of delivery and nutritional status were collected immediately after birth. Neonates were followed up for the first 72 hours to check clinical manifestations and evaluate EOI. Neonates who did not have clinical symptoms of infection did not admitted to neonatal care unit and clinically examined daily. Neonates with signs of suspected EOI (altered temperature, abnormal consciousness, seizures, shock, breath difficulty, apnea, vomiting, poor feeding, abdominal distention, abnormal heart rate, cyanosis, jaundice in the first 24 hours after birth, unexplained bleeding...) admitted to the neonatal unit, and screened for infection including blood cultures and count blood cells (WBC, Hb, PLT),blood glucose before use antibiotic (if indicated). The C-reactive protein (CRP) test is performed after 6 hours after birth. Blood count and CRP can be repeated after 24 hours.

Umbilical cord blood samples are collected immediately after birth. As soon as the newborn's umbilical cord was delayed clamping (except in cases need immediate resuscitation) and cut. Clean the umbilical cord area, take 2 ml of umbilical blood with a sterile syringe and put it in a sterile test tube containing lithium heparin anticoagulant and transfer it to the laboratory immediately afterwards. Quantitative cord blood procalcitonin level was performed by Cobas 6000/8000 analyzer, Roche Diagnostics, Japan. at the Laboratory Department, University of Medicine and Pharmacy Hospital, Hue University.

**Patient classification.** All newborns in this study were followed up for at least 72 hours after birth in the Obstetrics department and in the neonatal unit (if admitted) to assess for EOI. Based on clinical and paraclinical characteristics, neonates were categorized into two groups: EOI group and non-EOI group. Non-EOI group comprised: (1) asymptomatic neonates who were not hospitalized and (2) neonates with suspected infection who were hospitalized but improved without antibiotics or were discontinued antibiotics within 48 hours. In the latter case, improvement was confirmed by resolution of initial symptoms, absence of new signs of infection and negative blood culture and CRP test.

EOI Group: Neonates with clinical and laboratory evidence of infection were treated with antibiotics for a minimum of 7–10 days, following NICE 2021 guidelines [13]. Cord blood procalcitonin levels were not considered a diagnostic criteria for infection.

## Statistical analysis

The statistical analysis was done using SPSS version 20.0. The characteristics of the study population was analysed by percentages for quantitive variables and median for quantitive variables. The Chi-square test was used to compare variables between EOI and non-EOI group. The ROC curve was performed to detemine the threhold of PCT value for predicting EOI and its sensitivity (Se), specificity (Sp), positive predictive value (PPV), and negative predictive value (NPV). P-values of less than 0.05 was considered statistical significant.

## Ethical statements

The study was approved by the Institutional Review Board of the University of Medicine and Pharmacy, Hue University, Hue City, Vietnam (No. H2023/195, dated May 24, 2023). Parents of neonates agreed to voluntary participation and the written informed consent (about study and used data in medical records) was obtained before enrollment.

## Results

A total of 126 neonates were included in this study. Of these, 44 were diagnosed with EOI and 82 neonates were non-EOI. The basic characteristics of study population and risk factors of EOI were presnted in Table 1. The EOI group demonstrated a significantly higher incidence of prematurity and low birth weight neonates compared to the non-EOI group (p<0.05). However, no statistically significant differences were found between the two groups in terms of gender, mode of birth, or nutritional status (p>0.05). However, the EOI group had a significantly lower average weight (2527.27±715.08 gram) and gestational age (35.82±3.05 weeks) compared to the non-EOI group (37.55±2.33 g and 37.9±2.4 weeks, respectively; p<0.05). Risk factors of infection included preterm birth before 37 weeks (40.5%), prolonged rupture of membranes (28.6%), and maternal fever (22.2%). The median cord blood PCT concentration was

**Table 1. Baseline characteristics of of the study group.**

| Characteristics | | EOI group N = 44 [n (%)] | Non-EOI group N = 82 [n (%)] | p- value |
|---|---|---|---|---|
| Gender | Male | 23 (52.3) | 46 (56.1) | 0.681 |
| | Female | 21 (47.7) | 36 (43.9) | |
| Gestational age (weeks) | <37 | 24 (54.5) | 27 (32.9) | 0.018 |
| | ≥ 37 | 20 (45.5) | 56 (67.1) | |
| | Mean ± SD | 35.82±3.05 | 37.55±2.33 | 0.002 |
| Birth weight (grams) | <2500 | 20 (45.5) | 20 (24.4) | 0.015 |
| | 2500–<4000 | 24 (54.5) | 61 (74.4) | |
| | ≥4000 | 0 (0.0) | 1 (1.2) | |
| | Mean ± SD (g) | 2527.27±715.08 | 2891.46±55582 | 0.004 |
| Nutritional status | SGA | 2 (4.5) | 8 (9.8) | 0.471 |
| | AGA | 42 (95.5) | 72 (87.8) | |
| | LGA | 0 (0.0) | 2 (2.1) | |
| Types of delivery | Vaginal | 22 (50.0) | 40 (48.8) | 1.000 |
| | Caesarean section | 22 (50.0) | 42 (51.2) | |
| Risk factors of EOI | Suspected or confirmed infection in another baby in the case of a multiple pregnancy | 1 (2.3) | 0 (0.0) | 0.349 |
| | Pre-term birth following spontaneous labour before 37 weeks' gestation. | 24 (54.5) | 27 (32.9) | 0.018 |
| | Confirmed rupture of membranes for more than 18 hours before a pre-term birth | 11 (25.0) | 8 (9.8) | 0.023 |
| | Confirmed prelabour rupture of membranes at term for more than 24 hours before the onset of labour | 3 (6.8) | 15(18.3) | 0.079 |
| | Intrapartum fever higher than 38˚C if there is suspected or confirmed bacterial infection | 14 (31.8) | 15 (18.3) | 0.086 |
| | Clinical diagnosis of chorioamnionitis | 0 (0.0) | 1 (1.2) | 1.000 |
| | Spontaneous rupture of membranes between 12–18 hours prior to delivery | 4 (9.1) | 10 (12.2) | 0.769 |
| PCT value (ng/ml) in total population | Median | 0.154 | 0.097 | <0.001 |
| | (25th- 75th) | (0.092–0.197) | (0.082–0.134) | |
| PCT value (ng/ml) in preterm group | Median | 0.162 | 0.101 | <0.01 |
| | (25th- 75th) | (0.133–0.197) | (0.084–0.139) | |
| PCT value (ng/ml) in term group | Median | 0.157 | 0.096 | <0.01 |
| | (25th- 75th) | (0.099–0.377) | (0.079–0.133) | |

SGA, Small for gestational age; AGA, Appropriate for gestational age; LGA, Large for gestational age; EOI: Early-onset infection.

significantly higher in the EOI group 0.154 ng/ml (0.092–0.197) compared to the non-EOI group (0.097 ng/ml (0.082–0.134); p<0.001).

The ROC curve determined the optimal threshold value of PCT of 0.142 ng/ml with an AUC 0.751 (95% CI: 0.661–0.841, p<0.001) in the total population (Fig 1). At this cut-off, the Se, Sp, PPV, and NPV were 68.2%, 76.8%, 61.2%, and 81.8%, respectively. This result demostrated a good diagnosis accuracy of umbilical cord PCT for EOI.

We also performed a seperate analysis the ability to identify EOI for PCT in preterm and term group to minimize the involvement of gestational age in PCT concentration (Fig 2). In the preterm group, ROC analysis showed PCT presented a good performance in identifying EOI. The optimal cut-off value for preterm neonates was 0.122 ng/ml (AUC: 0.785, 95% CI: 0.658–0.911, p<0.001) corresponding a Se of 79.2%, Sp of 74.1%, PPV of 73.1%, and NPV of 80.0%. In term group, the optimal cut-off value was 0.150 ng/ml (AUC: 0.726, 95% CI: 0.583–0.860, p<0.01), with a Se of 60.0%, Sp of 80.4%, PPV of 52.2%, and NPV of 84.9%.

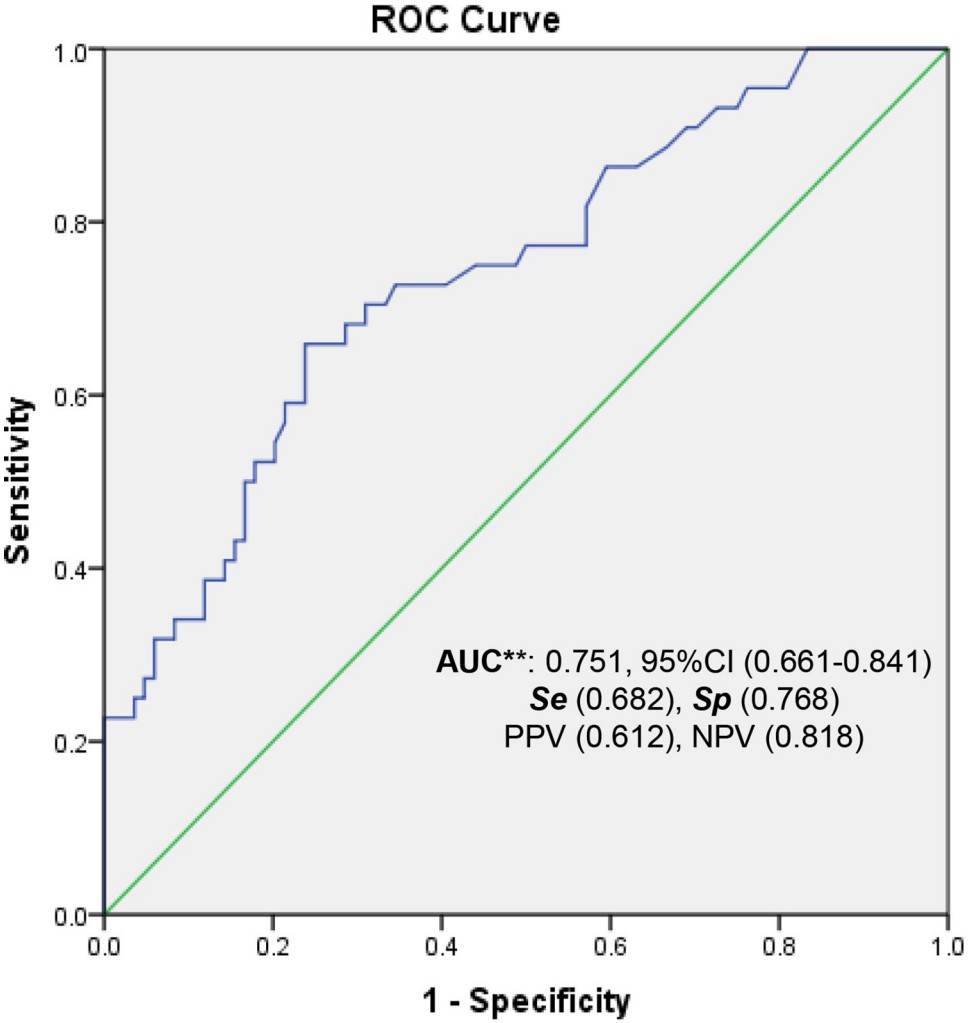

**Fig 1. ROC curve analysis of PCT for predicting early-onset neonatal infection in the total study population.**
ROC, receiver operating characteristic; AUC, area under the curve; Se, Sensitivity; Sp, Specificity; PPV, positive predictive value; NPV, negative predictive value; CI, Confidence Interval. **, *p*<0.001.

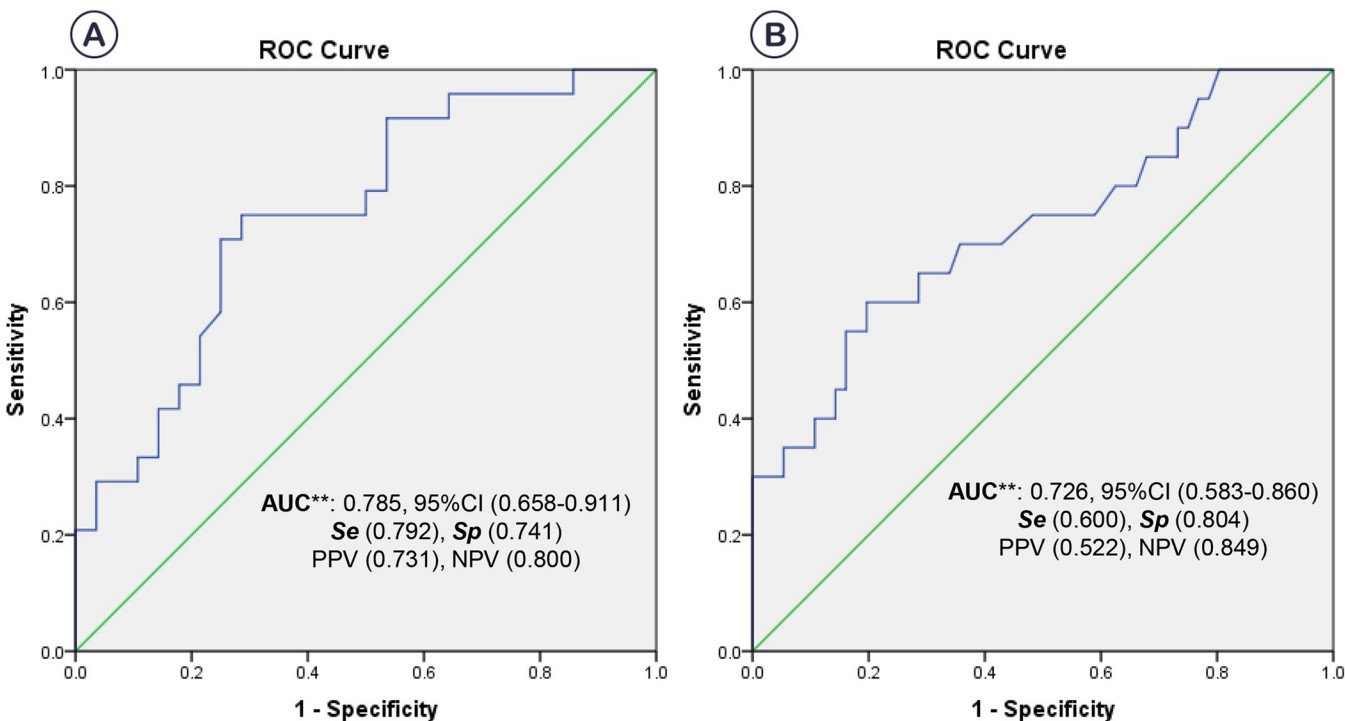

**Fig 2.** ROC curve analysis of PCT for predicting early-onset neonatal infection in preterm (A) and term group (B). ROC, receiver operating characteristic; AUC, area under the curve; Se, Sensitivity; Sp, Specificity; PPV, positive predictive value; NPV, negative predictive value; CI, Confidence Interval. **, $p < 0.001$.

## Discussion

EOI in newborns is an infection with an origin in the mother. Pathogens are transmitted from mother to child in utero or during labor. Therefore, exposure to pathogens trigger an inflammatory response in the fetus, increasing levels of pro-inflammatory cytokines like interleukin-6, interleukin-10 [14–16]. These cytokines, along with interleukin-1β, Tumor Necrosis Factor-α, and endotoxins, stimulate PCT production in the fetus [17]. A previous study demonstrated that maternal PCT levels during birth were not predictive of neonatal infection [18] while umbilical cord blood PCT concentrations have been shown to correlate with infectious intra-uterine [18, 19], suggesting its potential as a promising diagnosis biomarker for EOI.

In agreement with previous studies, our result showed that not all newborns with infection risk factors develop EOI. Therefore, it is possible that umbilical cord PCT could significantly support the early diagnosis of infection and minimize unnecessary empiric antibiotic use in cases suspected of EOI in NICUs. Additionally, Meena et al. reported that umbilical cord blood PCT was a good biomarker for identifying proven sepsis that showed better sensitivity and specificity values in comparison to peripheral blood PCT [20].

Our findings revealed that there was a statistically significant difference in cord blood PCT levels between the EOI and non-EOI groups, with the EOI group presenting significantly higher median PCT levels in total population and in each term or preterm neonates. The result on increased umbilical cord blood PCT concentrations in infected neonates compared to non-infected neonates was also reported in previous studies, such as the studies by Dongen O.R.E. et al. (2021) [10], Joram et al. (2011) [11], and Kordek et al. (2006) [18]. In our study, a PCT value of 0.142 was determined as the optimal cut-off for predicting EOI in the total population, with a Se of 68.2%, Sp of 76.8%, PPV of 61.2%, and NPV of 81.8%. We found different

diagnostic value of umbilical cord PCT based on sensitivity and specificity, PPV, NPV among previous studies [7, 11] and it remains unconclusive an optimal cutoff point to diagnosis EOI. For example, Joram et al. found that a cord blood PCT cut-off of 0.6 ng/ml had an AUC of 0.96 (95% CI: 0.95–0.98) for early EOI diagnosis in neonates (36.9 ± 3.8 weeks gestational age, 2783 ± 843.8 grams) with Se, Sp, NPV, and PPV values of 92.0%, 97.0%, 28.0%, and 99.0%, respectively [11]. Similarly, Kordek et al. (2006) reported a Se of 0.80 and an NPV of 0.95 for a cord blood PCT cut-off of 1.22 ng/ml [18]. Uyen's study determined that 0.23 ng/ml was the optimal cord blood PCT cut-off with an AUC of 0.87, following Se, Sp, PPV, NPV of 59.1%, 98.7%, 86.2%, and 94%, respectively [21]. Although there were differences in cut-off points among studies, these findings consist of the potential of umbilical cord PCT as a reliable bio-marker for early assessment of neonatal infections.

Although the proportion of EOI has decreased markedly in recent years, a more appropri-ate diagnostic approach has been challenged to minimizing empiric antibiotics for newborns with risk factors or suspected symptoms after birth. It is given that antibiotics can interfere with the development of the normal gut microbiota and increasing the opportunity of patho-genic bacterial colonization, and consequently affecting metabolism and immunity, especially in preterm newborns. In preterm neonates, who are at risk of infection, and often present with clinical signs that can be confused with normal characteristics due to gestational age or EOI, such as respiratory distress and poor feeding. Therefore, a separate analysis of PCT values con-sidering gestational age, especially in preterm neonates is essential to identify infected cases. Our study identified a PCT cut-off of 0.122 ng/ml for preterm group and 0.150 ng/ml for term group to diagnose EOI. Previous studies have reported a wide range of PCT cut-off values. For extremely preterm infants, Frerot et al. found that a PCT concentration >0.5 ng/mL was asso-ciated with a higher risk of EOS (OR 2.18; 95% CI 1.58–3.02; p < 0.0001) and the optimal cut-off was determined to be 0.7 ng/mL, with an AUC of 0.75 (Se 69%, Sp 70%) [22]. Joram at al reported an optimal cut-off of 0.6 ng/ml for both preterm and term group [11]. On the other hand, Dongen et al. revealed a lower cut-off of 0.1 ng/ml for neonates ≥ 32 weeks GA, with a Se of 83% and Sp of 62%. However, there could improve Sp to 95% but reduced Se to 50% at the cut-off of 0.6 ng/ml in the same study [10].

Although cord blood procalcitonin (PCT) has been investigated as a biomarker for early-onset infection (EOI), its diagnostic performance is not yet optimal. However, regarding inflammatory markers in cord blood, Su et al. reported that PCT can be a reliable marker to confirm or exclude EOI [19], Huetz et al. (2020) [2] and Hue-Bigé et al [23] demonstrated that PCT potentially guiding antibiotic therapy in neonates with risk factors or suspected clinical signs of infection. By applying a threshold of 0.6ng/ml or higher of umbilical cord blood PCT in algorithm to decide antibiotic prescription for EOI, Huetz et al. found that there was signifi-cantly reduction in antibiotic exposure in non-infected neonates by 39% [2].

Our study remains several limitations. Firstly, this was as a single hospital prospective study with a relatively sample size, therefore the result analysis is certain limited. However, as a hos-ptal with coordination between obstetrics and pediatrics in practicing, we obtained the neona-tal information and had close monitoring. Secondly, the lack of proven infection (cases based on positive blood cultures), all cases in EOI group were classified as probable sepsis. Therefore, further prospective studies with expanded study population is nescessary to obtain more evi-dence for recommendations about umbilical cord PCT.

## Conclusions

Umbilical cord PCT concentration can be an early diagnostic and differential marker of early neonatal sepsis.

## Supporting information

**S1 Data set.**
(CSV)

## Author Contributions

**Conceptualization:** Thi Thanh Binh Nguyen, Diep Anh Truong Thi, Quang Vinh Truong.

**Data curation:** Thi Thanh Binh Nguyen, Diep Anh Truong Thi, Quang Vinh Truong, Thi Ny Pham.

**Formal analysis:** Thi Thanh Binh Nguyen, Diep Anh Truong Thi.

**Funding acquisition:** Thi Thanh Binh Nguyen, Quang Vinh Truong, Thi Ny Pham.

**Investigation:** Thi Thanh Binh Nguyen, Diep Anh Truong Thi, Quang Vinh Truong, Thi Ny Pham.

**Methodology:** Thi Thanh Binh Nguyen, Diep Anh Truong Thi, Quang Vinh Truong, Thi Ny Pham.

**Project administration:** Thi Thanh Binh Nguyen, Diep Anh Truong Thi, Thi Ny Pham.

**Resources:** Thi Thanh Binh Nguyen, Quang Vinh Truong, Thi Ny Pham.

**Software:** Thi Thanh Binh Nguyen.

**Supervision:** Thi Thanh Binh Nguyen.

**Validation:** Thi Thanh Binh Nguyen.

**Visualization:** Thi Thanh Binh Nguyen.

**Writing – original draft:** Thi Thanh Binh Nguyen, Diep Anh Truong Thi, Quang Vinh Truong, Thi Ny Pham.

**Writing – review & editing:** Thi Thanh Binh Nguyen.

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
