## [Decision Letter · Decision Letter 0]

8 Dec 2024

PONE-D-24-51453The Diagnostic Accuracy of Umbilical Cord Procalcitonin in Predicting Early-Onset Neonatal Infection

PLOS ONE

Dear Dr. Nguyen,

Thank you for submitting your manuscript to PLOS ONE. After careful consideration, we feel that it has merit but does not fully meet PLOS ONE’s publication criteria as it currently stands. Therefore, we invite you to submit a revised version of the manuscript that addresses the points raised during the review process.

We look forward to receiving your revised manuscript.

Kind regards,

Benjamin M. Liu, MBBS, PhD, D(ABMM), MB(ASCP)

Academic Editor

PLOS ONE

Journal Requirements:

2. Thank you for stating the following financial disclosure: “This work was supported by research funds from Hue University (DHH 2023 – 04–202).”

6. We note that there is identifying data in the Supporting Information file <data.csv>. Due to the inclusion of these potentially identifying data, we have removed this file from your file inventory. Prior to sharing human research participant data, authors should consult with an ethics committee to ensure data are shared in accordance with participant consent and all applicable local laws. Data sharing should never compromise participant privacy. It is therefore not appropriate to publicly share personally identifiable data on human research participants. The following are examples of data that should not be shared: -Name, initials, physical address -Ages more specific than whole numbers -Internet protocol (IP) address -Specific dates (birth dates, death dates, examination dates, etc.) -Contact information such as phone number or email address -Location data -ID numbers that seem specific (long numbers, include initials, titled “Hospital ID”) rather than random (small numbers in numerical order) Data that are not directly identifying may also be inappropriate to share, as in combination they can become identifying. For example, data collected from a small group of participants, vulnerable populations, or private groups should not be shared if they involve indirect identifiers (such as sex, ethnicity, location, etc.) that may risk the identification of study participants. Additional guidance on preparing raw data for publication can be found in our Data Policy (https://journals.plos.org/plosone/s/data-availability#loc-human-research-participant-data-and-other-sensitive-data) and in the following article: http://www.bmj.com/content/340/bmj.c181.long. Please remove or anonymize all personal information (<specific identifying information in file to be removed>), ensure that the data shared are in accordance with participant consent, and re-upload a fully anonymized data set. Please note that spreadsheet columns with personal information must be removed and not hidden as all hidden columns will appear in the published file.

Additional Editor Comments:

Editor's comments:

1. PCT's role in other infections should be introduced and more references should be cited, with this one (PMID: 33337932) as an example (citing is optional)

2. "Early diagnosis is essential for optimizing treatment outcomes.": More references need to be cited, with these ones (PMID: 38042947 and 39221481) as examples (citing is optional)

3. "exposure to bacteria triggers an inflammatory response in the fetus, increasing levels of pro-inflammatory cytokines like interleukin-6, interleukin-10": In this statement bacteria should be changed to pathogens. More references should be cited, with this one (PMID: 38556084) as an example (citing is optional).

Reviewers' comments:

Reviewer's Responses to Questions

**Comments to the Author**

1. Is the manuscript technically sound, and do the data support the conclusions?

Reviewer #1: Partly

2. Has the statistical analysis been performed appropriately and rigorously? 

Reviewer #1: I Don't Know

3. Have the authors made all data underlying the findings in their manuscript fully available?

Reviewer #1: Yes

4. Is the manuscript presented in an intelligible fashion and written in standard English?

Reviewer #1: Yes

5. Review Comments to the Author

Reviewer #1: Although this research introduces a very important problem and may be a fetal one but, I don't think the manuscript added a new findings, as it is clear in the discussion part. Sample size needs to increase to give a reliable results

6. PLOS authors have the option to publish the peer review history of their article (what does this mean?). If published, this will include your full peer review and any attached files.

Reviewer #1: No

---

## [Author Response · Author response to Decision Letter 0]

17 Dec 2024

Respond to Editor and Reviewers

1. Comment 1. Please ensure that your manuscript meets PLOS ONE's style requirements, including those for file naming. The PLOS ONE style templates can be found at https://journals.plos.org/plosone/s/file?id=wjVg/PLOSOne_formatting_sample_main_body.pdf and https://journals.plos.org/plosone/s/file?id=ba62/PLOSOne_formatting_sample_title_authors_affiliations.pdf

Reply 1: Thank you for your suggestion. We have followed up the guideline for preparing manuscript. 

2. Comment 2. Thank you for stating the following financial disclosure: “This work was supported by research funds from Hue University (DHH 2023 – 04–202).”

Reply 2: Thank you for your suggestion. We have added this one in the rebuttal letter. 

This work was supported by research funds from Hue University, Viet Nam (DHH 2023 – 04–202). The funders had no role in study design, data collection and analysis, decision to publish, or preparation of the manuscript.

3. Comment 3: Please provide a complete Data Availability Statement in the submission form, ensuring you include all necessary access information or a reason for why you are unable to make your data freely accessible. If your research concerns only data provided within your submission, please write "All data are in the manuscript and/or supporting information files" as your Data Availability Statement.

Reply 3: Thank you for your suggestion. We have added this one in the manuscripts.

Reply 4: As data consists in the manuscript, it can be shared publicly. We have deposited data into the Supporting Information.

Reply 5: We have thoroughly reviewed and updated. 

6. We note that there is identifying data in the Supporting Information file <data.csv>. Due to the inclusion of these potentially identifying data, we have removed this file from your file inventory. Prior to sharing human research participant data, authors should consult with an ethics committee to ensure data are shared in accordance with participant consent and all applicable local laws. Data sharing should never compromise participant privacy. It is therefore not appropriate to publicly share personally identifiable data on human research participants. The following are examples of data that should not be shared: -Name, initials, physical address -Ages more specific than whole numbers -Internet protocol (IP) address -Specific dates (birth dates, death dates, examination dates, etc.) -Contact information such as phone number or email address -Location data -ID numbers that seem specific (long numbers, include initials, titled “Hospital ID”) rather than random (small numbers in numerical order) Data that are not directly identifying may also be inappropriate to share, as in combination they can become identifying. For example, data collected from a small group of participants, vulnerable populations, or private groups should not be shared if they involve indirect identifiers (such as sex, ethnicity, location, etc.) that may risk the identification of study participants. Additional guidance on preparing raw data for publication can be found in our Data Policy (https://journals.plos.org/plosone/s/data-availability#loc-human-research-participant-data-and-other-sensitive-data) and in the following article: http://www.bmj.com/content/340/bmj.c181.long. Please remove or anonymize all personal information (<specific identifying information in file to be removed>), ensure that the data shared are in accordance with participant consent, and re-upload a fully anonymized data set. Please note that spreadsheet columns with personal information must be removed and not hidden as all hidden columns will appear in the published file.

Reply 6: Thank you for your critical point regarding data sharing. We have thoroughly reviewed all variables and data in the Excel file. There is no sensitive information that can be identified, as explained above.

Additional Editor Comments:

Editor's comments:

Comment 1. PCT's role in other infections should be introduced and more references should be cited, with this one (PMID: 33337932) as an example (citing is optional).

Reply 1: Thank you for your suggestion. We added this information into the introduction of the revised manuscript and provided the following citations you recommended. 

“Previous studies have shown that, PCT is an early inflammatory marker with high specificity in response to severe systemic infections. It can be used for early detection, assessment of severity, and prognosis of the diseases.” 

7. Abdollahi A, Shoar S, Nayyeri F, Shariat M. Diagnostic Value of Simultaneous Measurement of Procalcitonin, Interleukin-6 and hs-CRP in Prediction of Early-Onset Neonatal Sepsis. Mediterr J Hematol Infect Dis. 2012;4(1):e2012028.

8. Liu BM, Hill HR. Role of Host Immune and Inflammatory Responses in COVID-19 Cases with Underlying Primary Immunodeficiency: A Review. J Interferon Cytokine Res. 2020;40(12):549-54.

Comment 2. "Early diagnosis is essential for optimizing treatment outcomes.": More references need to be cited, with these ones (PMID: 38042947 and 39221481) as examples (citing is optional).

Reply 2: Thank you for your suggestion. We cited the following references for this point in the revised manuscript.

2. Huetz N, Launay E, Gascoin G, Leboucher B, Savagner C, Muller JB, et al. Potential impact of umbilical-cord-blood procalcitonin-based algorithm on antibiotics exposure in neonates with suspected early-onset sepsis. Frontiers in pediatrics. 2020;8:127.

3. Liu BM. Epidemiological and clinical overview of the 2024 Oropouche virus disease outbreaks, an emerging/re-emerging neurotropic arboviral disease and global public health threat. J Med Virol. 2024;96(9):e29897.

4. Liu BM, Mulkey SB, Campos JM, DeBiasi RL. Laboratory diagnosis of CNS infections in children due to emerging and re-emerging neurotropic viruses. Pediatr Res. 2024;95(2):543-50.

Comment 3. "exposure to bacteria triggers an inflammatory response in the fetus, increasing levels of pro-inflammatory cytokines like interleukin-6, interleukin-10": In this statement bacteria should be changed to pathogens. More references should be cited, with this one (PMID: 38556084) as an example (citing is optional).

Reply 3: Thank you. We edited this part in the revised manuscript and cited the following references: 

14. Mangogna A, Agostinis C, Ricci G, Romano F, Bulla R. Overview of procalcitonin in pregnancy and in pre-eclampsia. Clin Exp Immunol. 2019;198(1):37-46.

15. Liu BM, Li NL, Wang R, Li X, Li ZA, Marion TN, et al. Key roles for phosphorylation and the Coiled-coil domain in TRIM56-mediated positive regulation of TLR3-TRIF-dependent innate immunity. J Biol Chem. 2024;300(5):107249.

Reviewers' comments:

Reviewer's Responses to Questions

Comments to the Author

1. Is the manuscript technically sound, and do the data support the conclusions?

Reviewer #1: Partly

Reply 1: Thank you. Our study aimed to provide more evidence to support recommendations on using umbilical cord PCT for early EOI diagnosis, addressing the current lack of consensus in the literature.

About sample size, we also mentioned our limitations in this study. However, this study was conducted over a period of more than one year. During this time, we followed nearly 2765 newborns born in our hospital, monitoring all risk factors and early-onset infections. From this cohort, we obtained a sample size of 126 cases, including 44 cases of EOI. 

To provide further clarity on the research process and data collection, we included this information in the methods section of the revised manuscript.

2. Has the statistical analysis been performed appropriately and rigorously?

Reviewer #1: I Don't Know

Reply 2: Thank you. We have carefully rechecked our data and statistical analysis. We have confirmed the accuracy of our original findings and will retain the current results.

5. Review Comments to the Author

Reviewer #1: Although this research introduces a very important problem and may be a fetal one but, I don't think the manuscript added a new findings, as it is clear in the discussion part. Sample size needs to increase to give a reliable results.

Reply 3: We are very grateful for the supportive feedback on our manuscript and its research content. We hope to have the opportunity to publish our research findings to contribute more evidence for the early diagnosis of neonatal early-onset infection, a challenge that persists in neonatology, and make a substantial contribution to the neonatal field.

Regarding sample size, we also mentioned our limitations in this study. However, this study was conducted over a period of more than one year. During this time, we followed nearly 2765newborns born in our hospital, monitoring all risk factors and early-onset infections. From this cohort, we obtained a sample size of 126 cases, including 44 cases of EOI.

---

## [Editor Report · Decision Letter 1]

20 Dec 2024

The Diagnostic Accuracy of Umbilical Cord Procalcitonin in Predicting Early-Onset Neonatal Infection

PONE-D-24-51453R1

Dear Dr. Nguyen,

We’re pleased to inform you that your manuscript has been judged scientifically suitable for publication and will be formally accepted for publication once it meets all outstanding technical requirements.

Kind regards,

Benjamin M. Liu, MBBS, PhD, D(ABMM), MB(ASCP)

Academic Editor

PLOS ONE
---

## [Editor Report · Acceptance letter]

7 Jan 2025

PONE-D-24-51453R1 

PLOS ONE

Dear Dr. Nguyen, 

I'm pleased to inform you that your manuscript has been deemed suitable for publication in PLOS ONE. Congratulations! Your manuscript is now being handed over to our production team.

Kind regards, 

on behalf of

Dr. Benjamin M. Liu 

Academic Editor

PLOS ONE